# Lightweight Vision Transformers for Low Energy Edge Inference

Shashank Nag, Logan Liberty, Aishwarya Sivakumar, Neeraja J. Yadwadkar, Lizy K. John

*Chandra Family Department of Electrical and Computer Engineering*

*The University of Texas at Austin*

Austin, TX, USA

Emails : {*shashanknag, loganliberty, rathna.siva02*}@utexas.edu, *neeraja@austin.utexas.edu, ljohn@ece.utexas.edu*

*Abstract*—**Vision Transformer models have been performing increasingly well in recent times. However, their computational demands make them infeasible to be deployed on edge devices with latency and energy constraints. Weightless Neural Networks (WNNs) are look-up-table based models, that are different from conventional Deep Neural Networks, and offer a low-latency, low-energy alternative. In this work, we seek to combine aspects of vision transformers and weightless neural networks to design Lightweight Vision Transformers that are efficient for edge inferences - to strike a desirable trade-off between the hardware requirements of transformers and accuracy achieved. We analyze the I-ViT-T vision transformer variant to observe that roughly 57% of the computations are within the Multi Layer Perceptron (MLP) layers. We estimate the hardware savings in replacing these layers with our proposed weightless layers, and evaluate such models for accuracy. Preliminary results with the I-ViT-T model suggest that the weightless layers introduced in place of the MLP layers result in a significant speedup for a lower hardware resource requirement, as compared to a systolic array based accelerator implementation for the MLPs. When evaluated on end-to-end performance, this model variant offers a 2.9x drop in energy per inference over the baseline model – at the cost of about 6% drop in model accuracy on the CIFAR-10 dataset. We continue our efforts to improve the model accuracy and extend this work to larger transformer variants and benchmarks, while trying to optimize the hardware resource consumption.**

## I. INTRODUCTION

With the advent of computer vision in several applications, there has been an increasing demand for low-latency machine learning inferences at the edge. In recent times, transformer-based vision models have been gaining popularity in image classification tasks [1]–[3], and are state-of-the-art in some datasets. However, the large and complex nature of these models pose significant challenges in their deployment, particularly in terms of hardware and energy resources required, and high inference latency [4]. This makes them inefficient for edge inferences.

We see an opportunity by tapping into Weightless Neural Networks (WNNs) that are a class of Look-Up-Table (LUT)-based neural networks designed specially to be efficiently deployed on edge FPGA devices, with the design matching the underlying logic fabric on the FPGA. However, despite their advantages of low latency and power, WNNs are limited in their learning capability, and do not perform well beyond smaller datasets, such as MNIST [5]–[7].

In this work, we identify key features of vision transformers and WNNs that suggest combining aspects of these, and seek to design Quasi-Weightless Vision Transformer models. In doing so, we propose a class of models that offer model performance advantages of transformers, while offering hardware performance benefits of WNNs - often resulting in a tradeoff between the two. We note that low energy requirements, ultra-low latency and a LUT based implementation makes weightless models excellent candidates for deployment on edge FPGAs for inference tasks. Hence we particularly target edge FPGA devices for hardware evaluations of these models against their corresponding baselines.

The remainder of the paper is organized as follows. In Section II we discuss the background and motivation behind the work, and identify specific features of Vision Transformers and Weightless Neural Networks that make their fusion worthwhile. In Section III we analyze these models and discuss the proposed design, and in Section IV we evaluate the said proposed design. We discuss the limitations and future work in Section V and Section VI concludes the paper.

## II. BACKGROUND & MOTIVATION

### A. Vision Transformers

Vision Transformers are encoder-based models that are inspired from the transformer models designed for language tasks. As shown in Fig. 1, these involve self-attention applied to a series of tokens generated by splitting the image into patches. In case of an image classification task, a classification head is attached to the sequence, and extracted at the end of the encoder stack to get the predicted class. Prior studies have explored quantization techniques to make these models more efficient for hardware deployment. MobileViT [8] combines aspects of CNNs and vision transformers to offer a mobile-friendly vision transformer. I-ViT [9] proposes a fully int-8 quantized vision transformer variant, that is particularly suitable for inference on dedicated hardware.

In this work, we seek to explore avenues to make vision transformers further light-weight, that are complementary to such existing techniques.

This research was supported in part by Semiconductor Research Corporation (SRC) Task 3148.001, National Science Foundation (NSF) Grant #2326894, and NVIDIA Applied Research Accelerator Program Grant. Any opinions, findings, conclusions, or recommendations are those of the authors and not of the funding agencies.

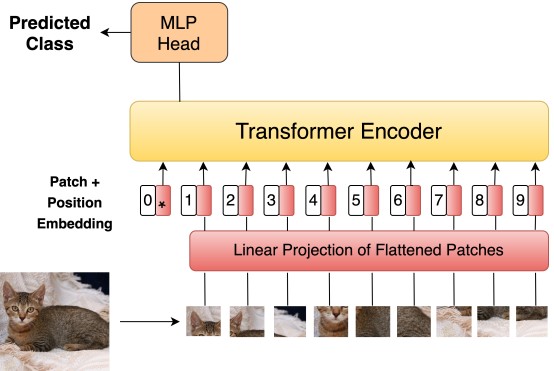

Fig. 1. A typical vision transformer model. The model typically consists of a stack of encoder blocks. Figure adapted from [1]

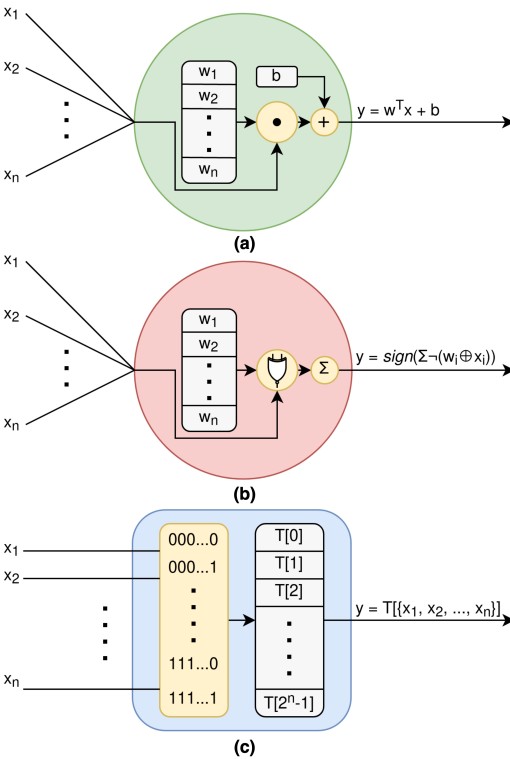

Fig. 2. (a) Conventional Neuron : Each neuron multiplies inputs with weights and adds them. (b) Binary Neural Network Neuron : The weights being binary, the multiplication operation is substituted by a XNOR (c) Weightless Neuron : In contrast, the input sequence is "looked up" in the LUT with no MAC operations involved

## B. Weightless Neural Networks

Weightless Neural Networks are neural networks inspired by the dendritic trees of biological neurons. These involve the idea of avoiding intensive computations in neural networks, by replacing the regular weight-based neural layers with "weightless" layers. These weightless layers are primarily comprised of look-up table (LUT) based neurons, and eliminate the need for power and resource hungry multiply-accumulate operations in conventional neurons (Fig. 2). While earlier works in this field [10], [11] suffered from high memory requirements, recent works like BTHOWeN [6] & ULEEN [5] have demonstrated significantly lower compute and memory requirements compared to iso-accuracy MLPs & CNNs for image classification tasks. This has sparked a renewed interest in WNN research and potential usage. While WNNs suffer from a higher training complexity, their LUT-based architecture makes them well-suited for deployment on FPGAs (that have underlying LUT slices) for inferences. We note that since we primarily aim to design an inference-efficient model, the training complexity isn't a major concern.

## C. Combining Weightless Neural Networks and Vision Transformers

As mentioned earlier, vision transformers incur high resource utilization, high latency, and involve several power-hungry multiply-accumulate operations. On the contrary, WNNs offer a low-latency energy efficient solution involving look-up operations, resulting in low resource utilization. Furthermore, while the model sizes of vision transformers scale quadratically with latent dimensions, we could potentially scale it linearly with WNNs. By incorporating self-attention, vision transformers are able to learn positional dependence in images and perform well on larger datasets – the current WNNs struggle in doing so. Moreover, most layers in vision transformers are linear-like, and from prior work [5], we know that WNNs have been able to learn patterns represented by linear layers more efficiently. Thus, all of these factors suggest that many of the aspects of vision transformers and WNNs are complementary to each other, and the question of how to combine these is worth studying.

In this work, we focus on identifying specific layers of the vision transformer models that are compute-intensive, and aim at replacing such layers with weightless layers to provide hardware benefits. We note that transformer models have repeated blocks of a few layers stacked one over the other. As a consequence of the repeated blocks, we need to modify intermediate layers of a larger network to introduce weightless layer. Such a modification hasn't been explored in the past, to the best of our knowledge. In this regard, we redesign and adapt weightless layers that preserve the dimensions of the latent space of transformer network, and integrate them together to design a Quasi-Weightless Vision Transformer model.

We also note that while there are alternatives to vision transformers for vision tasks, transformers are ubiquitous in the realm of language models. An effective solution of integrating vision transformers and WNNs would pave the way for future exploration of extending these concepts to language models. Vision transformers are architecturally similar to language models, and at the same time, these have smaller datasets and model sizes compared to language models. Thus, this study also serves as a good stepping stone towards eventually studying weightless language models.

## III. QUASI-WEIGHTLESS VISION TRANSFORMERS

### A. Initial Thoughts & Key Insights

For our analysis and implementation, we use the I-ViT vision transformer model [9] shown in Fig. 3 as our baseline. We profiled the I-ViT-T model variant to analyze the computational requirements of the layers within it. Based on this analysis, we figured out that the MLP layers are quite compute intensive, and perform about 57% of total MAC (multiply and accumulate) operations within each encoder layer. In addition, the MLP layers are also found to be responsible for about 66.7% of the total model weights within the encoder layer. These findings could be generalized to other variants of vision transformers as well, as shown in [4]. Additionally, prior studies indicate that the WNN layers can out-perform iso-accurate MLP layers in terms of reduced latency and reduced energy consumption [5], [6].

Thus, we plan to replace the MLP layers of the vision transformer with similarly designed fully connected (FC) weightless layers. We expect that this change would result in both reduced latency and reduced energy consumption, at a minimal drop in accuracy. Note that since the LUTs can inherently learn non-linearities, we would not require additional non-linear activation layers. By doing so, we hope to gain the benefits of the energy savings from WNNs combined with the performance advantages brought forth by the self-attention layers of the transformers.

### B. System Design

We design a modified multi-layer weightless network similar to the one in [5] and [12] to replace the MLP layer as discussed in Section III-A. After adding two layers of LUTs, we include a summation layer that sums responses from the LUTs in the previous layer, in order to generate integer outputs for the final layer. Prior work on WNNs mostly involved discriminator-based models that had these layers generate scores for different output classes. In contrast, we preserve the latent space dimensionality since these layers form intermediate parts of a larger network. In order to ensure that the introduced weightless network mimics the way a MLP layer processes each token independently, we flatten the input activations along the row, and pass them to these layers one row at a time.

Fig. 4 shows the proposed design of the quasi-weightless vision transformer. We integrate the weightless layer into each of the encoders of the vision transformer model, with the original MLP layer being replaced by it.

## IV. EVALUATION

### A. Experimental Setup

To evaluate and iterate upon our design, we adopt the following methodology :

1) Setup a baseline model, and design the proposed quasi-weightless model replacing MLPs with configurable LUT-based weightless layers for the same model variant.
2) Train model, tweak hyperparams, report accuracy.

3) Analytically estimate hardware (HW) resource consumption, energy, and performance.
4) Iterate over the above steps 1-3 incorporating feedback from these analytical reports.

After careful consideration, we identify the I-ViT-T model [9] (int8 quantized variant) as our baseline for experimentation. This is primarily due to the fact that we found full-precision DeiT [2] models to be infeasible to train with larger batch sizes when integrated with the weightless network. Furthermore, as noted in Sec. III-B, since the weightless network generates integer outputs, it would be more compatible with I-ViT. Having chosen the baseline model, we redefine the PyTorch model for its quasi-weightless version. We consider the CIFAR-10 [13] image classification task as our benchmark for evaluation, as this is a commonly used dataset for edge application evaluations. We pick the ImageNet [14] pre-trained vision transformers, modify these model layers, and fine-tune them on CIFAR-10. We performed various iterations over weightless layer configurations, training strategies, and other hyperparameters - this is still an ongoing effort.

In order to provide estimates of hardware performance improvements when deployed on a FPGA, we require a synthesized design for both the MLP layers (that we are replacing) and for the weightless layers. The synthesized designs for MLP will serve as our baseline to compare the performance of the WNN layers against. We consider the PYNQ-Z1 FPGA board, a typical edge device, as the target device for obtaining the hardware utilization reports of our synthesized design.

The MLP layer that we intend to replace, consists of the Dense→ShiftGELU→Dense sections shown in Fig. 3. The Dense bubble represents a GeMM operation (general matrix multiplication) while the the ShiftGELU operation performs an integer approximation of the sigmoid function used for non-linear activation in ViTs [9]. It consists of bitwise logical and arithmetic operations, along with floating point division and multiplication operations. For the chosen I-ViT-T model, the matrix multiplications in the MLP layer would be 1) $196 \times 192$ by $192 \times 768$ and 2) $192 \times 768$ by $768 \times 192$. We designed a systolic array based accelerator, typically used in literature for MLP accelerators [15] to consider hardware estimates for our baseline. The systolic architecture designed utilizes a pipelined blocking matrix multiplication method. To provide more points of comparison against the resources, latency, and energy consumption of the quasi-weightless model we synthesized several different sizes of systolic arrays: 4x4, 8x8, 16x16, and 32x32. We also designed a high-throughput inference accelerator for the introduced weightless layers, similar to the one proposed in ULEEN [5] and DWN [12], and synthesized the same on the target FPGA.

In addition to the systolic arrays, we also created a dedicated processing unit for computing the ShiftGELU and ShiftMAX operations. ShiftMax implements an integer version of the typical ViT SoftMax, which translates attention scores into probabilities [9]. Similar to ShiftGELU, ShiftMax consists of few fundamental arithmetic computations and other simpler bitwise operations such as shift.

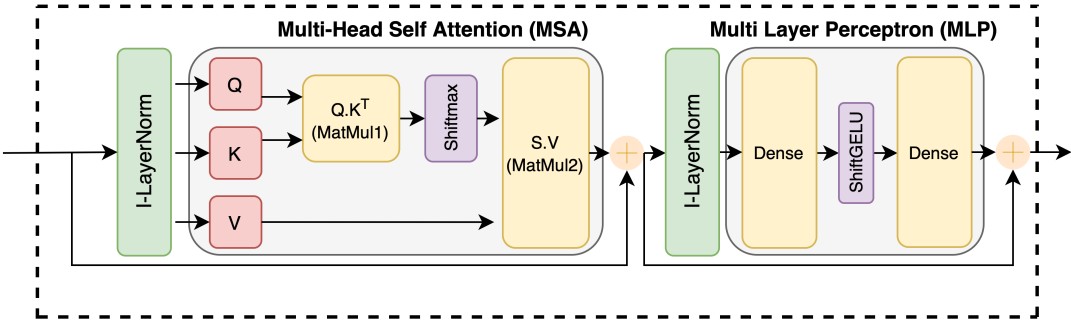

Fig. 3. An encoder layer in I-ViT - Integer Vision Transformer [9]. Each of the intermediate operations are performed in int8 precision, and the resultant int32 values are quantized back to int8 using a scale factor. Shiftmax and ShiftGELU represent element-wise non-linear operations. The full I-ViT model has multiple of these encoder layers stacked.

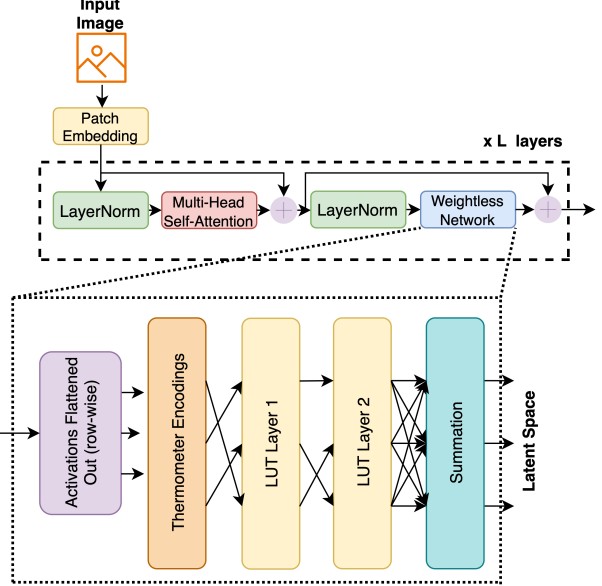

Fig. 4. Proposed Quasi-Weightless Vision Transformer

| Method | LUTs | FFs | 1/Throughput | Energy/Sample |
|---|---|---|---|---|
| 4x4 Systolic | 1,435 | 744 | 1,806,336 Cycles | 307.077 uJ |
| 8x8 Systolic | 5,601 | 3,051 | 460,800 Cycles | 188.928 uJ |
| 16x16 Systolic | 22,337 | 12,363 | 119,808 Cycles | 101.837 uJ |
| 32x32 Systolic | 91,937 | 50,385 | 32,256 Cycles | 62.2541 uJ |
| Weightless Network Implementation | 15,543 | 20,755 | 196 Cycles | 1.99 uJ |

TABLE II
HARDWARE ARCHITECTURE PERFORMANCE COMPARISON USING A SINGLE GeMM (DENSE) (196x192)*(192x768) WITHIN THE MLP LAYER

| Stage | Baseline Model | Quasi-Weightless Model |
|---|---|---|
| Q, K, V | 15.5635 uJ (each) | 15.5635 uJ (each) |
| MatMul 1 | 18.1574 uJ | 18.1574 uJ |
| ShiftMax | 10.196 uJ | 10.196 uJ |
| MatMul 2 | 18.1574 uJ | 18.1574 uJ |
| MLP Dense 1 | 62.2541 uJ | |
| ShiftGELU | 60.505 uJ | 1.99 uJ |
| MLP Dense 2 | 62.2541 uJ | |
| **Total** | **278.215 uJ** | **95.1913 uJ** |

TABLE III
ENERGY CONSUMPTION COMPARISON BETWEEN METHODS FOR A SINGLE LAYER (PER SAMPLE INFERENCE). A 32x32 SYSTOLIC ARRAY IS USED FOR THE NON-WEIGHTLESS LAYERS.

## B. Results

| Model Variant | Baseline Model | Quasi-Weightless Model |
|---|---|---|
| I-ViT-T [9] | 95.41% | 89.44% |
| DeiT-T [2] | 87.1% | 80.3% |

TABLE I
MODEL ACCURACY COMPARISONS OF THE BASELINE VISION TRANSFORMER MODELS VS. THE DESIGNED QUASI-WEIGHTLESS MODELS ON THE CIFAR-10 DATASET

We evaluate the model accuracy of the baseline model and the quasi-weightless model for the I-ViT-T model variant. While we noted the issues surrounding implementation with DeiT models in the previous section, we still include an evaluation of the same to serve as an additional comparison. [1] As noted in Table I, with CIFAR-10, the baseline I-ViT model accuracy was found to be 95.41%, while the weightless model was able to achieve an accuracy of 89.44%. On the DeiT-T model, the baseline model was 87.1% accurate versus the quasi-weightless model being 80.3% accurate.

These preliminary findings suggest that the quasi-weightless

---

[1] The original works on I-ViT [9] and DeiT [2] do not report model accuracies on CIFAR-10 for the Tiny (T) variants. In order to setup and report baseline model accuracies, we follow the exact same methodology and code implementations provided by the authors. We ascertain that our implementation is as intended by training the DeiT-B model on the CIFAR-10 dataset. With this setup, we are able to reproduce the accuracy of 97.5% when trained on CIFAR-10 from scratch, and 99.1% when finetuned on ImageNet pretrained model as reported in [2], within 0.1% & 0.2% error margin respectively.

model accuracy is comparably close to the baseline. We also note that these weightless models that have been integrated with vision transformers perform much better compared to prior weightless networks, that report the accuracy on CIFAR-10 in the range of 50% [5].

Making a direct comparison between the performance of a single MLP Dense layer implementation with the weightless network is difficult as the introduced weightless network replaces both the MLP Dense layers and the ShiftGELU layer. As shown in Table II, for a single MLP Dense layer the systolic architecture was found to consume significantly higher energy to process each sample compared to the weightless network implementation. Of the different sized systolic array based matrix multipliers, the 32x32 implementation consumed the least amount of energy at 62.2541 uJ per MLP Dense layer and took 32,256 cycles to complete a single layer. In contrast, the weightless network that replaces the entire MLP block consumed 1.99 uJ of energy and took 196 cycles to process a sample – this is a significant performance uplift over the systolic architecture.

We also performed estimations for end-to-end energy consumption per sample inference for the baseline implementation, as well as the quasi-weightless model implementation, for a single encoder in the stack. We considered the 32x32 systolic array architecture for this comparison, and considered using the same architecture as the original model for the parts of the quasi-weightless model implementation that remain unchanged. In doing so, the energy associated with the Multi-head self-attention (MSA) layers remain unchanged. Table III summarizes these results, and shows the quasi-weightless model implementation outperforming the baseline model implementation by a significant degree. The layers with the highest energy consumption (MLP Dense 1, ShiftGELU, MLP Dense 2) are particularly the ones replaced with the weightless network, and hence provides the most uplift in energy consumption. The total end-to-end energy consumption for a single sample with the baseline model implementation is 278.215 uJ and with the quasi-weightless model is 95.1913 uJ - which is a 65.79% increase in energy efficiency. We also note that by eliminating about 66.7% of the overall model weights (from the MLP layers), we also save a corresponding amount of memory. While we account for BRAM access energy in our estimates, typically a lot of these model weights are stored off-chip and are fetched on-demand. The resultant savings in reduced weight movement to on-chip memory would be much more pronounced.

These preliminary results on performance and power consumption lend support to the potential of employing weightless models for edge applications.

## V. LIMITATIONS / FUTURE DIRECTIONS

Our more immediate improvements revolve around the limitations with the current accuracy we were able to achieve, as well as the comparisons we could fairly make. Given we have sub 90% accuracy, we will continue optimizing the weightless layers while ensuring we still meet the resource constraints of the target PYNQ-Z1 FPGA. We will also be performing further comparisons for a few other datasets, such as CIFAR-100 and ImageNet. We note that the current model design is not quite efficient in terms of training time - thereby restricting our experiments to limited benchmarks and smaller model variants. Improved training techniques is something that we would definitely like to focus on. We would finally like to perform a more end-to-end deployment of the model for performing a more robust comparison between the quasi-weightless model and the baseline model. Our current comparisons are primarily analytical estimates.

Aside from these improvements, we also hope to extend this work in the future by incorporating learnable mapping techniques [12] and knowledge distillation from the MLP to the weightless network. We would also like to explore if the learnable mapping techniques or alternate bit concatenations result in improved training times. In terms of the transformer models, we would like to evaluate additional model variants such as iVIT-B and iVIT-S, as well as other benchmarks. Added comparison against other binarized models, such as BinaryViT [16] and BiViT [17] would also be useful to prove the effectiveness of quasi-weightless model over binarized models. In terms of the future model design itself, we eventually hope to extend to weightless self-attention layers to create a fully weightless vision transformer. This would eventually pave the way for decoder-only and encoder-decoder models, to target Large Language Models.

## VI. CONCLUSION

Quasi-weightless vision transformers appear to be a promising lightweight alternative to traditional vision transformers based on our end-to-end analysis of the energy savings & model accuracy. They achieve significantly higher accuracy than previous weightless neural networks as well. The weightless layers that replace the MLP layers in these quasi-weightless vision transformers show a significant improvement in sample latency and energy consumption compared to traditional systolic array based architectures, and the results are indeed positive. We report an overall 66.7% reduction in the required model weights, and a 2.9x savings in end-to-end energy when comparing the quasi-weightless model implementation to the baseline design, traded off by a $\sim 6\%$ drop in accuracy on CIFAR-10 benchmark. We also eliminated the need for BRAMs and off-chip weight movement for the MLP layers that are replaced by the weightless layers. However, we note that the amount of LUTs required for a single weightless layer is still notably large. Since these can't be shared across the encoder, we are left with the challenging task of fitting multiple layers onto a single FPGA. Reductions can possibly be made to the size of the weightless layers without a massive impact in the performance, and we intend to research this further. By reducing the overall size of the weightless layers, we hope to fit multiple layers onto a single FPGA. We continue our efforts towards better model designs and training techniques in order to close the gap in

model accuracy between the quasi-weightless models and the baselines.

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
