# OpenReview forum: "Lightweight Vision Transformers for Low Energy Edge Inference"
_iscaconf.org/ISCA/2024/Workshop/MLArchSys — MLArchSys 2024 OralPoster_

### Official Review · Reviewer_tb3V · 2024-05-26
**Promising preliminary results for inference energy and throughput of ViTs**

**Confidence:** 3
**Rating:** 7

**Detailed Feedback And Questions For Authors:**

* Why *model size of ViTs scale quadratically with latent dimensions while WNNs scale linearly?*
* What is your analytical model for performance estimation?

**Top Reasons To Accept The Paper:**

* Explores using LUTs for FPGA implementation of MLPs in vision transformers, showing significant energy reductions without a significant drop in accuracy
* Thorough and interesting plan for future work

**Top Reasons To Reject The Paper:**

* Accuracy is only evaluated on CIFAR, which at this point is a toy dataset.
* Comparison with further quantization and distillation methods is missing.

---

### Official Review · Reviewer_ETp4 · 2024-05-26
**Review for "Lightweight Vision Transformer for Low Energy Edge Inference"**

**Confidence:** 3
**Rating:** 6

**Detailed Feedback And Questions For Authors:**

* Please explain how the WNN scales with the scaling of the ViT base model? How would the accuracy compare considering different model sizes (or matmul dimensions)?
* In Table II, it's not clear what the second and third columns are.
* Please provide data to compare the cost of training for the modified model, in comparison with the baseline.

**Top Reasons To Accept The Paper:**

* The paper proposes a novel idea to replace Feedforward network in an encoder-based ViT model with Weightless Neural Network to reduce energy and improve performance. The initial evaluation shows promising results, with a dip in accuracy (6%) in exchange for reducing latency and energy by 2.9x, making the model suitable for inference on edge devices and deployed on FPGAs.
* The technique is well justified and the paper provides convincing data on how to modify the ViT models to make them quasi-weightless.

**Top Reasons To Reject The Paper:**

* The initial results shows the technique is promising, however, higher accuracy is needed to make this a viable solution
* Training WNNs could be more complicated and more costly than the ViT models. There is not much discussion and exploration on the training approach. It's mainly postponed to future work. In Section IV, they mention the training approach by fine-tuning after modifying the ImageNet model. It is worth comparing the complexity and cost of the fine-tuning process.
* As the ViT model scales to larger more powerful models, are the LUT components easily scalable as well? How is the accuracy compared in larger networks? Also, the FF networks are sharded based on well known approaches such as [Shoeybi, et.al. Megatron-LM], however, it's not clear to me how the LUTs are scaled.
* The paper mentions the linear scaling of LUTs with the scaling of latent dimensions, in comparison with quadratic scaling of matmuls.  This doesn't seem accurate. Increasing the latent dimension, will increase the activation size quadratically. This will heavily impact the attention blocks (which are not replaced in the new model). It looks to me that both FFN layers and LUTs would be impacted the same way, with increasing the latent dimensions.

---

### Official Review · Reviewer_kabj · 2024-05-29

**Confidence:** 3
**Rating:** 5

**Detailed Feedback And Questions For Authors:**

The paper introduces a novel approach by combining Vision Transformers (ViTs) with Weightless Neural Networks (WNNs) for efficient edge inference, but several weaknesses are notable:
* **Dataset Limitations**: Evaluation is restricted to the CIFAR-10 dataset, limiting the understanding of the model's potential for more complex tasks. Including larger datasets like CIFAR-100 and ImageNet would provide better insights into its generalization capabilities.
* **Accuracy Drop**: The ~6% accuracy drop on CIFAR-10 raises concerns about the model's practicality in real-world applications where accuracy is critical.
* **Lack of Detailed Implementation**: Insufficient details on the implementation of weightless layers and training procedures hinder reproducibility. More comprehensive information on the training setup and hyperparameters is needed.
* **Comparison with Other Techniques**: The paper lacks comparisons with other model compression techniques such as pruning, quantization, and distillation, which are necessary to contextualize the efficiency gains and strengthen the claims.

**Top Reasons To Accept The Paper:**

* The paper is well-written, with effective use of figures and tables to illustrate the architecture and results, enhancing the clarity and comprehensibility of the presented research.
* The proposed method achieves significant energy savings and reduced latency, addressing critical challenges in deploying deep learning models on resource-constrained devices.

**Top Reasons To Reject The Paper:**

* The paper's evaluation is restricted to the CIFAR-10 dataset, which does not adequately demonstrate the model's performance on more complex, real-world tasks.
* The observed ~6% accuracy drop on CIFAR-10 is a major concern, undermining the practicality of the model for real-world applications where maintaining high accuracy is critical.
* The lack of detailed information on the implementation of weightless layers and training procedures hinders reproducibility, making it difficult for other researchers to replicate the results.

---

### Official Review · Reviewer_4ms1 · 2024-05-30
**Review for Quasi-Weightless ViT paper**

**Confidence:** 5
**Rating:** 7

**Detailed Feedback And Questions For Authors:**

Summary
The paper proposes a Lightweight Vision Transformer called "Quasi-Weightless ViT," which replaces the MLP layers in the baseline I-VIT vision transformer model with weightless fully connected layers of the same size. This approach reduces overall energy consumption by 34.2%, decreases required weights by 66.7%, and improves throughput while maintaining comparable accuracy (with only ~6% accuracy loss). The paper is well-written and clear. I recommend its acceptance.


Pros
	1. Understanding of Trade-offs: The authors demonstrate a clear understanding of the advantages and disadvantages of traditional approaches, effectively leveraging strengths to address challenges. They incorporate WNNs within Vision Transformers, addressing key challenges and leading to more efficient model (Quasi-Weightless ViT).
	2. Energy Efficiency and Latency: The model achieves 65.79% energy efficiency, a 66.7% reduction in weight footprint, and significantly reduced latency compared to systolic array-based implementations.
	3. Addressing Limitations: The authors acknowledge the limitations of their proposed model and have a clear plan to address them.
	4. Accuracy and Deployment Trade-off: The paper discusses the trade-off between accuracy and deployment performance, providing a balanced view of the model's practical implications.
	5. Edge Device Applicability: The exploration of Vision Transformers on edge devices is a significant research point, and the technique proposed is promising.

Cons:
	1. Training Time: The proposed model has longer training times. The authors plan to address this in future versions, but immediate suggestions for mitigation could be helpful.
	2. Resource Utilization on PYNQ-Z1 FPGA: Detailed resource utilization relative to the entire device's resources would provide more insights.
	3. DSP Utilization: The DSP results in Table II need clarification. It should be clear whether DSPs results were ignored or if the authors enforced (restricted) Vivado tool to use LUT only instead of DSPs.
	4. End-to-End Results: The paper lacks end-to-end actual implementation results, relying on analytical results. Detailed implementation plans or intermediate results could enhance credibility. Keep in mind the cost of the interconnections between the digital blocks on the programmable logic (PL) and other blocks.
	5. Complex Dataset Testing: The model was tested on the CIFAR-10 dataset, which is relatively simple. Testing on more complex datasets is recommended to validate real-world performance.
	6. Tables Representations: The tables could be better organized and presented.
	7. Analytical Model: The analytical model used to calculate throughput and latency for systolic arrays should be included for clarity.

**Top Reasons To Accept The Paper:**

1. Understanding of Trade-offs: The authors demonstrate a clear understanding of the advantages and disadvantages of traditional approaches, effectively leveraging strengths to address challenges. They incorporate WNNs within Vision Transformers, addressing key challenges and leading to more efficient model (Quasi-Weightless ViT).
	2. Energy Efficiency and Latency: The model achieves 65.79% energy efficiency, a 66.7% reduction in weight footprint, and significantly reduced latency compared to systolic array-based implementations.
	3. Addressing Limitations: The authors acknowledge the limitations of their proposed model and have a clear plan to address them.
	4. Accuracy and Deployment Trade-off: The paper discusses the trade-off between accuracy and deployment performance, providing a balanced view of the model's practical implications.
Edge Device Applicability: The exploration of Vision Transformers on edge devices is a significant research point, and the technique proposed is promising.

**Top Reasons To Reject The Paper:**

None

---

### Decision · Program_Chairs · 2024-05-30

**Decision:**

Accept (Oral/Poster)

**Comment:**

Congratulations! We are pleased to inform you that your paper has been accepted for presentation at MLArchSys 2024. We look forward to your participation at the workshop. Further details regarding the schedule and format will be provided soon. See you at the workshop!